# Characterization of *Pseudofusicoccum* Species from Diseased Plantation-Grown *Acacia mangium*, *Eucalyptus* spp., and *Pinus massoniana* in Southern China

**DOI:** 10.3390/pathogens12040574

**Published:** 2023-04-08

**Authors:** Guoqing Li, Wenxia Wu, Linqin Lu, Bingyin Chen, Shuaifei Chen

**Affiliations:** Research Institute of Fast-Growing Trees (RIFT), Chinese Academy of Forestry (CAF), Zhanjiang 524022, China

**Keywords:** *Botryosphaeriales*, fungal pathogen, virulence, phylogeny

## Abstract

Fungi from *Pseudofusicoccum* (*Phyllostictaceae, Botryosphaeriales*) have been reported as pathogens, endophytes, or saprophytes from various woody plants in different countries. Recently, *Botryosphaeriales* isolates were obtained from the dead twigs of *Acacia mangium*, *Eucalyptus* spp., *Pinus massoniana*, and *Cunninghamia lanceolata* in Guangdong, Guangxi, Hainan, and Fujian Provinces in southern China. This study aimed to understand the diversity, distribution, and virulence of these *Pseudofusicoccum* species on these trees. A total of 126 *Pseudofusicoccum* isolates were obtained, and the incidences of *Pseudofusicoccum* (percentage of trees that yielded *Pseudofusicoccum*) on *A*. *mangium*, *P*. *massoniana*, *Eucalyptus* spp., and *C. lanceolata* were 21%, 2.6%, 0.5%, and 0%, respectively. Based on the internal transcribed spacer (ITS), translation elongation factor 1-alpha (*tef1*), and β-tubulin (*tub2*) loci, 75% of the total isolates were identified as *P*. *kimberleyense*, and the remaining isolates were identified as *P*. *violaceum*. For *P*. *kimberleyense*, the majority of isolates (83%) were from *A*. *mangium*, and the rest were from *P*. *massoniana* (14%) and *Eucalyptus* spp. (3%). Similarly, the proportion of isolates of *P*. *violaceum* from *A*. *mangium*, *P*. *massoniana*, and *Eucalyptus* spp. were 84%, 13%, and 3%, respectively. Inoculation trials showed that the two species produced expected lesions on the tested seedlings of *A*. *mangium*, *E*. *urophylla* × *E*. *grandis*, and *P*. *elliottii*. This study provides fundamental information on *Pseudofusicoccum* associated with diseases in main plantations in southern China.

## 1. Introduction

The genus *Pseudofusicoccum* was proposed in 2006 based on DNA sequence data to accommodate ‘*Fusicoccum stromaticum*’ [1,2]. The status has been revised several times in recent years, and now, it is classified into *Phyllostictaceae* of *Botryosphaeriales* [3,4]. To date, nine species have been included in the genus [5]. As pathogens, endophytes, or saprophytes, species of *Pseudofusicoccum* have been reported from many woody plants, such as *Mangifera indica*, *Acacia synchronica*, and *Eucalyptus* spp., in countries including Australia, Brazil, India, South Africa, Thailand, Uruguay, and Venezuela [6]. The main diseases associated with these fungi include die-back, stem canker, and fruit rot [7,8,9].

Large plantations have been established in China, benefiting from a series of forestry programs [10]. In the subtropical and tropical areas of the country, more than 11 Mha of *Cuninghamia lanceolata*, 8 Mha of *Pinus massoniana*, and 5 Mha of *Eucalyptus* trees have been planted to date [11]. *Acacia mangium* is another popular species for plantations, but it has a relatively limited cultivation area [12].

In recent years, many diseases have been reported from these plantation trees in China, and numerous pathogens have been reported, including the fungi of *Botryosphaeriaceae*, *Calonectria*, *Ceratocystis*, *Cryphonectriaceae*, *Mycosphaerellaceae*, *Quambalaria*, and *Teratosphaeriaceae*, and the bacteria *Ralstonia solanacearum* [13,14,15,16]. Out of these, more than 20 species in *Botryosphaeriales* have been detected, and most of them reside in the genera *Botryosphaeria*, *Diplodia*, *Lasiodiplodia*, and *Neofusicoccum* of *Botryosphaeriaceae* [15,17,18], but *Pseudofusicoccum* has not been reported in the country to date.

In 2020, disease surveys were conducted in plantations of *A*. *mangium*, *C*. *lanceolata*, *Eucalyptus* spp., and *P*. *massoniana* trees in southern China. Symptomatic branches presenting die-back caused by *Botryosphaeriales* fungi were collected, and *Pseudofusicoccum*-like isolates were isolated from these hosts. This study aimed to: (1) identify the species of these *Pseudofusicoccum* isolates from *A*. *mangium*, *C. lanceolata*, *Eucalyptus* spp., and *P*. *massoniana*; (2) determine their geographic distribution on these four different hosts; and (3) evaluate their virulence on *A*. *mangium*, *E*. *urophylla* × *E*. *grandis*, and *P*. *elliottii* trees.

## 2. Materials and Methods

### 2.1. Sample Collection and Fungal Isolation

Disease surveys were conducted in adjacent plantations of *A*. *mangium*, *Eucalyptus* spp., *P*. *massoniana*, and *C*. *lanceolata* in Guangdong, Guangxi, Hainan, and Fujian Provinces in southern China. Die-back of trees occurred commonly in these plantations. A total of 16 sites, 3–5 sites for each province, were selected for sample collection. At each site, about 50 trees with diseased symptoms for each host were selected, and one branch with dead twigs was collected from each diseased tree. *Acacia mangium* and *Eucalyptus* spp. trees were approximately 3–4 years old, and *P*. *massoniana* and *C*. *lanceolata* trees were approximately 7–8 years old. Branches with dead tips were cut off with a high tree pruner. 

*Botryosphaeriales*-like fungi were isolated, and pure cultures were obtained, as described by Li et al. [17]. For branches with pycnidium, the pycnidium was transferred to the medium using a sterile steel needle. For branches without pycnidium, small pieces from the inner part of the branch were transferred to the medium using a sterile scalpel. Four pycnidia or cuttings from different positions on the branch were transferred to one 2% malt extract agar (MEA) (20 g melt extract powder and 20 g agar dissolved in 1 L of water) plate, and one *Botryosphaeriales*-like isolate for each branch was selected for further study. All of the cultures were deposited in the Culture Collection (CSF) of the Research Institute of Fast-growing Trees (RIFT), Chinese Academy of Forestry (CAF), Zhanjiang, Guangdong Province, China.

### 2.2. DNA Extraction, PCR Amplification, and Sequencing

The total genomic DNA of the isolate was extracted from the mycelium of 7-day-old cultures, grown on MEA at 25 °C in the dark, using the CTAB method [19]. A total of 2 μL RNase A (10 mg/mL) was added to each DNA sample and samples were incubated at 37°C for 1 h to remove RNA. DNA samples were checked for quality and concentration using a NanoDrop 2000 Spectrometer (Thermo Fisher Scientific Inc., Waltham, MA, USA). For PCR amplification, the DNA samples were diluted to approximately 100 ng/µL with DNase/RNase-free ddH_2_O (Sangon Biotech Co., Ltd., Shanghai, China).

The internal transcribed spacer (ITS) was amplified using the ITS1 and ITS4 primers [20]. Translation elongation factor 1-alpha (*tef1*) was amplified using the EF1F and EF2R primers [21]. β-tubulin (*tub2*) was amplified using the BT-2a and BT-2b primers [22]. The PCR reaction mixture contained 35 μL of total volume, which consisted of 18 μL 2× High Fidelity PCR Master Mix (mixture of Super-Fidelity DNA Polymerase, MgCl_2_, dNTP Mix) (Sangon Biotech Co., Ltd., Shanghai, China), 1 μL of each forward and reverse primers, 13 μL ddH_2_O, and 2 μL DNA. The amplification conditions were as follows: an initial denaturation step at 94 °C for 3 min, 35 cycles of 94 °C for 1 min, 55 °C for ITS and *tub2*, 59 °C for *tef1* for 1 min, and 72 °C for 1 min, and a final elongation step at 72 °C for 10 min.

The PCR reactions were conducted in a thermocycler (BIO-RAD T100^TM^, Bio-Rad Laboratories, Inc., Hercules, CA, USA). The PCR products were examined by electrophoresis in 1.5% agarose gel with 4SGelred (Sangon Biotech Co., Ltd., Shanghai, China) 1× Tris-acetate-EDTA (TAE) buffer at a constant voltage (80 V) for 40 min and visualized under UV light using a Molecular Imager Gel Doc^TM^ XR System (Bio-Rad Laboratories, Inc., California, USA). The PCR products were sequenced in both directions by the Beijing Genomics Institution, Guangzhou, China. Sequences were inspected and manually corrected in Geneious v. 9.1.4 [23]. All of the sequences generated in this study were submitted to GenBank (http://www.ncbi.nlm.nih.gov, accessed on 22 March 2023).

### 2.3. Phylogenetic Analyses

Sequences of ITS, *tef1*, and *tub2* were generated for all of the isolates obtained in this study. Based on the sequences of the three loci, the genotype of each isolate was determined, and 1–2 isolates were selected for phylogenetic analyses. Preliminary identification was conducted by sequence similarity searching using BLAST (https://blast.ncbi.nlm.nih.gov/Blast.cgi, accessed on 8 July 2022), and the available sequences of all of the species in *Pseudofusicoccum* containing ex-type isolates were downloaded from NCBI for phylogenetic analyses. The sequences were aligned using the online version of MAFFT v.7 (http://mafft.cbrc.jp/alignment/server/, accessed on 10 February 2023) [24], with the iterative refinement method (FFT-NS-i setting). The alignments were checked manually and edited in MEGA v.6.0.5 [25].

Phylogenetic analyses were conducted using maximum likelihood (ML), maximum parsimony (MP), and Bayesian inference (BI) methods for datasets of ITS, *tef1*, and *tub2*, and the combination of the three loci. ML analyses with 1000 bootstrap replicates were conducted with PhyML v.3.0 [26]. MP analyses were conducted with PAUP v.1.0b10 [27], and gaps were treated as a fifth character. BI analyses were performed with MrBayes v. 3.2.7a [28] on the CIPRES Science Gateway v. 3.3. For ML and BI analyses, the best-fit model of nucleotide substitution for each dataset was determined with jModelTest v.2.1.5 [29]. Bootstrap support values were evaluated using 1000 bootstrap replicates [30]. The phylogenetic analyses were rooted in *Botryosphaeria dothidea* (CBS 115476). The trees were visualized in FigTree v. 1.4.4.

### 2.4. Inoculation Trials

To determine the virulence of the species identified in this study, inoculation trials were conducted in a greenhouse using potted healthy seedlings of 1-year-old *A*. *mangium*, 1-year-old *E*. *urophylla* × *E*. *grandis*, and 2-year-old *P*. *elliottii* at the South China Experiment Nursery (SCEN), located in Zhanjiang, Guangdong Province, China. These seedlings were approximately 170 cm high and 2 cm in diameter at the root collar.

For each seedling, a wound (5 mm in diameter) was made on the stem (approximately 30 cm above the root collar) using a cork borer to remove the bark and expose the cambium, and the mycelial plug (5 mm diameter) from a 7-day-old culture of the selected isolate was placed into the wound with the mycelium facing the xylem. The wound with the mycelial plug was sealed with masking tape immediately to avoid contamination and desiccation. Negative control was conducted with a clean 2% MEA plug. Ten trees were inoculated for each isolate, including the negative controls. After one month, lesion lengths were measured and recorded. Re-isolations were made from the inoculated plants to fulfill Koch’s postulates. One-way analysis of variance (ANOVA) was used to determine the differences in virulence among isolates utilizing SPSS v. 20 [31].

## 3. Results

### 3.1. Fungal Isolation

A total of 500 samples were collected from *A*. *mangium*, 804 from *Eucalyptus* spp., 650 from *P*. *massoniana*, and 400 from *C*. *lanceolata* trees in southern China (Table 1). A total of 126 *Pseudofusicoccum* isolates identified based on ITS sequences were obtained from these trees (Table 1 and Table 2). Out of these, 105 isolates (83.3%) were obtained from *A*. *mangium*, 17 isolates (13.5%) were from *P*. *massoniana*, four isolates (3.2%) were from *Eucalyptus* spp., and no isolates were from *C*. *lanceolata*.

### 3.2. Phylogenetic Analyses and Species Identification

The ITS, *tef1*, and *tub2* loci were amplified for all 126 isolates (Table 2). The sequence fragments were approximately 520 bp for ITS, 280 bp for *tef1*, and 430 bp for *tub2*. Sequence alignments were deposited in TreeBASE (30240). Isolates from other studies used for phylogenetic analyses were shown in Table 3. According to the phylogenetic analyses of the ITS, *tef1*, *tub2*, and the combined datasets, the isolates in this study (Group A and Group B) were most closely related to *P*. *kimberleyense* and *P*. *violaceum* (Table 2). The sequence similarity of *P. kimberleyense* isolates in this study with the type of isolate (CMW 26156) were 99.42% to 99.81% for the ITS region, 98.35% to 99.01% for the *tef1* gene region, and 99.08% to 100% for the *tub2* gene region. The sequence similarity of *P. violaceum* isolates in this study with the type of isolate (CMW 22679) were 99.42% to 100% for the ITS region, 99.01% to 100% for the *tef1* gene region, and 99.54% to 100% for the *tub2* gene region. Although they also clustered or were closely related to *P*. *ardesiacum* and *P*. *africanum* based on the ITS dataset, they separated distinctly with the two species based on *tef1*, *tub2*, and combined datasets (Figure 1 and Appendix A). The ITS and *tub2* trees showed close relationships among the isolates in this study with species of *P*. *kimberleyense* and *P*. *violaceum*, and the *tef1* and combined trees provided clear results that separated isolates in Group A and Group B from the two known species (Figure 1 and Appendix A). Additionally, some isolates in this study formed an independent clade in the phylogenetic trees, but these clades had poor bootstrap values. Based on the phylogenetic analyses of the four datasets, isolates in Group A and Group B were considered the known species of *P*. *kimberleyense* and *P*. *violaceum*, respectively.

### 3.3. Distribution of Pseudofusicoccum

For the four plantation hosts, the incidence of *Pseudofusicoccum* (percentage of trees that yielded *Pseudofusicoccum*) was 21% on *A*. *mangium*, 2.6% on *P*. *massoniana*, 0.5% on *Eucalyptus* spp., and zero on *C*. *lanceolata* based on results in Table 1. Two *Pseudofusicoccum* species were identified from these trees, and *P*. *kimberleyense* was the dominant, comprising 75% of all of the obtained isolates, followed by *P*. *violaceum*. For isolates of *P*. *kimberleyense*, 83% were from *A*. *mangium*, 14% were from *P*. *massoniana*, and 3% were from *Eucalyptus* spp. For isolates of *P*. *violaceum*, 84% were from *A*. *mangium*, 13% were from *P*. *massoniana*, and 3% were from *Eucalyptus* spp. (Figure 2).

### 3.4. Inoculation Trials

For the two species identified, 1–3 isolates were selected for inoculations on each of the original hosts. Six isolates of the two species were used to inoculate *A*. *mangium* and *E*. *urophylla* × *E*. *grandis*, and four isolates were used to inoculate *P*. *elliottii* (Table 2). Typical lesions with a depression at the inoculation site were observed on inoculated plants, in comparison with wounds on the negative controls. Lesion and wound lengths were recorded one month after inoculation. The results showed that all of the isolates produced lesions on the tested plants, while the controls produced only small wound reactions (Figure 3 and Figure 4). The inoculated species were re-isolated from the lesions, but never from the negative controls.

Overall, the lengths of lesions caused by the inoculated isolates were similar to the wounds produced by the negative controls for each of the three tree species. On *A*. *mangium*, three isolates of the two species (*P*. *kimberleyense*: CSF18503 and CSF19318, *P*. *violaceum*: CSF19320) produced lesions significantly longer than the wounds caused by the controls, while the other three isolates produced lesions not significantly different from the wounds caused by the controls (*p* = 0.05) (Figure 4A). On *P*. *elliottii*, the inoculated isolates produced lesions not significantly different from the wounds caused by the controls, except for isolates CSF18491 (*P*. *kimberleyense*) and CSF18430 (*P*. *violaceum*) (Figure 4B). On *E*. *urophylla* × *E*. *grandis*, the inoculated isolates produced lesions significantly longer than the wounds in the negative controls, except for isolate CSF19067 (*P*. *kimberleyense*) (Figure 4C).

## 4. Discussion

In this study, 126 isolates of *Pseudofusicoccum* were obtained from the plantations of *A*. *mangium*, *Eucalyptus* spp., and *P*. *massoniana* from four provinces in southern China. Two species, *P*. *kimberleyense* and *P*. *violaceum*, were identified based on multi-phylogenetic analyses of ITS, *tef1*, and *tub2* loci. To our knowledge, this is the first report of *Pseudofusicoccum* species in China.

Genealogical concordance phylogenetic species recognition (GCPSR) provides criteria and has been applied for species delimitation for many years [39,40]. Multi-gene phylogenetic analyses without the morphological characteristics were used commonly for the identification of described species of *Botryosphaeriales*, including species of *Pseudofusicoccum* [41,42,43]. For *Pseudofusicoccum* species, the common loci used for phylogenetic analyses are ITS, *tef1*, and *tub2*, which can provide sufficient information to distinguish most species [2,5,32,44]. The phylogenetic analyses in this study revealed that trees based on each of the loci and a combination of the three loci were necessary for species identification, and *tef1* and combined datasets were more efficient in species delimitation in this genus.

Previous studies have detected *Pseudofusicoccum* species in various hosts in different countries [45,46]. Out of these, *P*. *kimberleyense* was first described on *Adansonia gibbosam*, *Acacia synchronica*, *Eucalyptus* sp., and *Ficus opposita* in Australia [32,47] and also reported from *Carya illinoinensis* in Brazil [48]. *Pseudofusicoccum violaceum*, first reported from *Pterocarpus angolensis* in South Africa [36], has been reported on *Tinospora cordifolia* in India [49] and *Mangifera indica* in Malaysia [50]. This study also showed that both were detected in *A*. *mangium*, *Eucalyptus* spp., and *P*. *massoniana*. A high proportion of isolates on *A*. *mangium*, compared with very rare ones on *Eucalyptus* spp. and *P*. *massoniana*, and no isolates on *C*. *lanceolata* in this study, revealed that species of *Pseudofusicoccum* associated with diseases may have a host preference in the environment.

Inoculation trials revealed that the two *Pseudofusicoccum* species identified in this study were virulent to the three tested hosts. This is consistent with previous studies showing that these species are also important pathogens to many hosts, including *Mangifera indica* [50,51,52], *Syzygium malaccense* [53], and *Artemisia annua* [9]. Although some isolates presented relatively weak virulence to hosts, such as *P*. *adansoniae*, *P*. *ardesicum*, and *P*. *kimberleyense* on baobab taproots [47], *P*. *africanum* on *Mimusops caffra* [33], and some *P*. *kimberleyense* and *P*. *violaceum* isolates presenting minor lesions on inoculated seedlings in this study, the co-occurrence with other botryosphaeriaceous fungi revealed that *Pseudofusicoccum* plays a role in disease occurrence and development [54].

The current study provides foundational data on the diversity, distribution, and virulence of *Pseudofusicoccum* from plantations of *A*. *mangium*, *Eucalyptus* spp., and *P*. *massoniana* in southern China. This study also provides evidence of the host preference of these agents. These *Pseudofusicoccum* species associated with stem canker and die-back indicate a new potential threat to these plantations and should not be ignored in disease management in the future.

## Figures and Tables

**Figure 1 pathogens-12-00574-f001:**
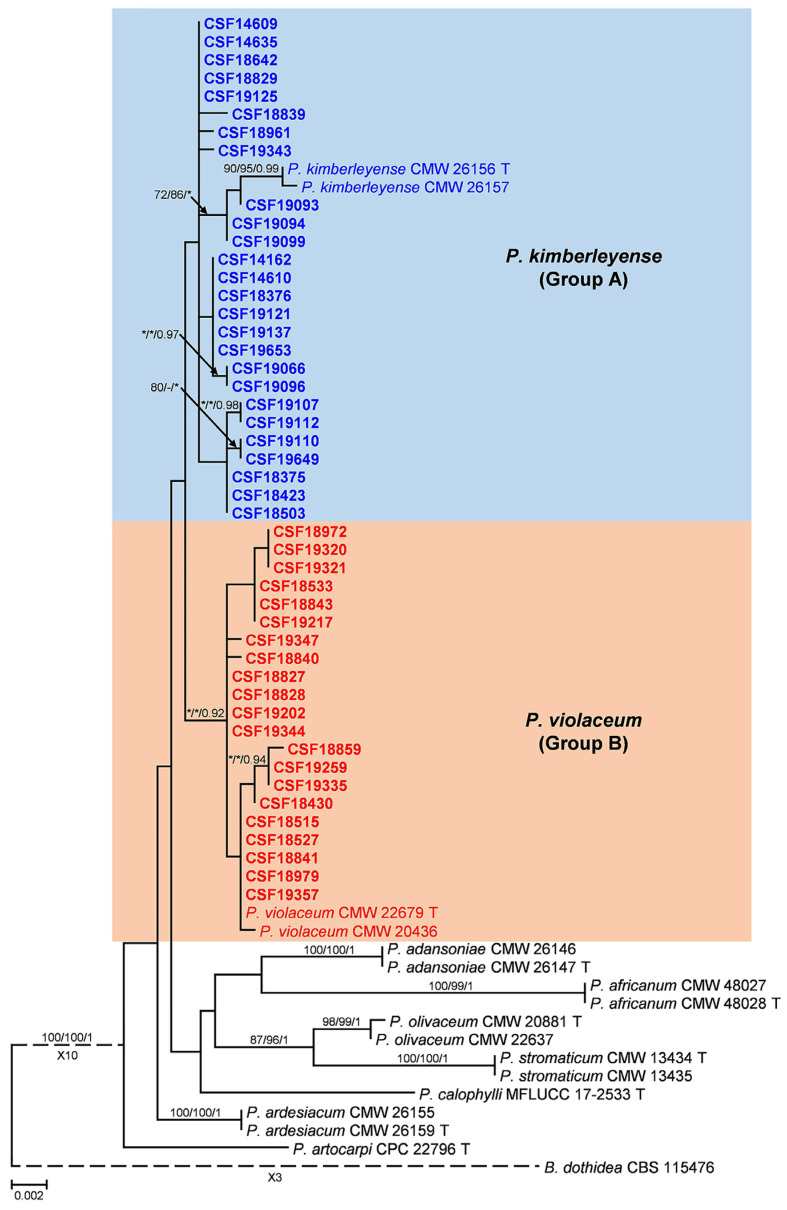
Phylogenetic tree based on maximum likelihood (ML) analyses of the combined DNA dataset of ITS, *tef1*, and *tub2* loci for *Pseudofusicoccum* species. Isolates in blue (Group A) and red (Group B) colors in bold were sequenced in this study. Bootstrap support values ≥ 70% for ML and MP (maximum parsimony) and probabilities values of BI (Bayesian inference) ≥ 0.9 are presented above the branches as follows: ML/MP/BI, bootstrap support values < 70% and probabilities values < 0.9 are marked with ‘*’, and absent are marked with ‘-’. Ex-type isolates are marked with ‘T’. The trees were rooted in *Botryosphaeria dothidea* (CBS 115476).

**Figure 2 pathogens-12-00574-f002:**
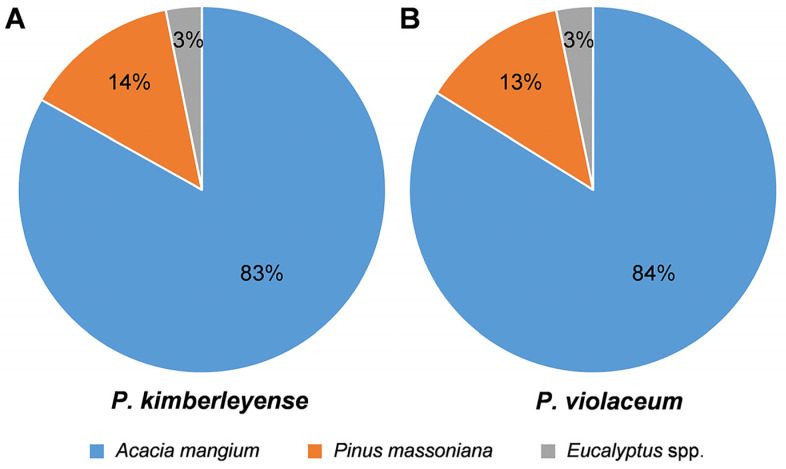
Percentage of isolates on *Acacia mangium*, *Pinus massoniana*, and *Eucalyptus* spp. for each species of *Pseudofusicoccum kimberleyense* (**A**) and *P*. *violaceum* (**B**).

**Figure 3 pathogens-12-00574-f003:**
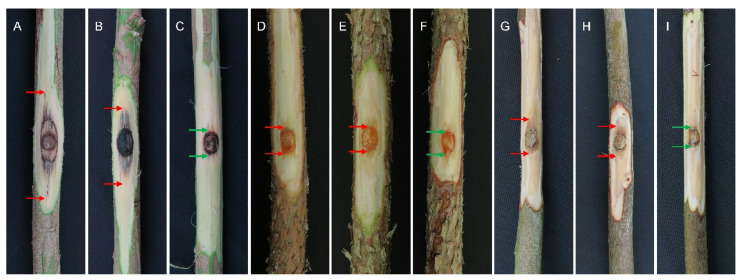
Symptoms observed on *Acacia mangium* (**A**–**C**), *Pinus elliottii* (**D**–**F**), and *Eucalyptus urophylla* × *E*. *grandis* (**G**–**I**) one month after inoculation. (**A**) Lesion produced by isolate CSF18503 (*P*. *kimberleyense*); (**B**) lesion produced by isolate CSF19320 (*P*. *violaceum*); (**C**) negative control; (**D**) lesion produced by isolate CSF18491 (*P*. *kimberleyense*); (**E**) lesion produced by isolate CSF19418 (*P*. *violaceum*); (**F**) negative control; (**G**) lesion produced by isolate CSF19064 (*P*. *kimberleyense*); (**H**) lesion produced by isolate CSF18895 (*P*. *violaceum*); (**I**) negative control.

**Figure 4 pathogens-12-00574-f004:**
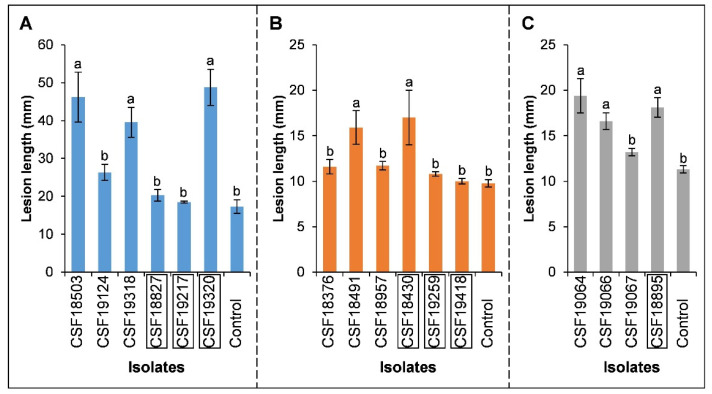
Column chart indicating the average lesion length (mm) produced by isolates of *Pseudofusicoccum* on the tested seedlings of *Acacia mangium* (**A**), *Pinus elliottii* (**B**), and *Eucalyptus urophylla* × *E*. *grandis* (**C**). Bars represent the standard error of the mean, and different letters on the bars indicate treatment means that are significantly different (*p* = 0.05). The isolates without boxes are *P*. *kimberleyense*, and the isolates with boxes are *P*. *violaceum*.

**Table 1 pathogens-12-00574-t001:** Samples collected and *Pseudofusicoccum* isolates obtained in this study.

Plantation Tree Species	Number of Samples				Number of *P. kimberleyense*/*P. violaceum* Isolates			
	Guangdong	Guangxi	Hainan	Fujian	Guangdong	Guangxi	Hainan	Fujian
*Acacia mangium*	150	100	200	50	9/0	20/10	35/6	15/10
*Pinus massoniana*	250	200	50	150	7/1	2/0	4/0	0/3
*Eucalyptus* spp.	254	200	200	150	0/0	0/0	3/1	0/0
*Cunninghamia lanceolata*	150	100	0	150	0/0	0/0	0/0	0/0

**Table 2 pathogens-12-00574-t002:** Isolates sequenced and used for phylogenetic analyses and inoculation trials in this study.

Species	Genotype ^a^	Isolation No. ^b^	Host	Location	GPS Information	Collector	GenBank Accession No. ^c^		
							ITS	*tef1*	*tub2*
*Pseudofusicoccum kimberleyense*	AAA	CSF14609 ^d^	*Acacia mangium*	Yangdong County, Yangjiang Region, Guangdong Province, China	22°01′27″ N, 112°11′17″ E	G.Q. Li	OQ659775	OQ659901	OQ660027
*P. kimberleyense*	AAA	CSF14635 ^d^	*Pinus massoniana*	Yangchun County, Yangjiang Region, Guangdong Province, China	21°55′31″ N, 111°38′37″ E	G.Q. Li	OQ659776	OQ659902	OQ660028
*P. kimberleyense*	AAA	CSF18370	*P. massoniana*	Huazhou County, Maoming Region, Guangdong Province, China	21°47′05″ N, 110°28′35″ E	G.Q. Li	OQ659777	OQ659903	OQ660029
*P. kimberleyense*	AAA	CSF18519	*A. mangium*	Beiliu County, Yulin Region, Guangxi Province, China	22°47′12″ N, 110°17′53″ E	G.Q. Li	OQ659778	OQ659904	OQ660030
*P. kimberleyense*	AAA	CSF18531	*A. mangium*	Beiliu County, Yulin Region, Guangxi Province, China	22°47′12″ N, 110°17′53″ E	G.Q. Li	OQ659779	OQ659905	OQ660031
*P. kimberleyense*	AAA	CSF18848	*A. mangium*	Shangsi County, Fangchenggang Region, Guangxi Province, China	22°06′50″ N, 107°52′60″ E	G.Q. Li	OQ659780	OQ659906	OQ660032
*P. kimberleyense*	AAA	CSF18860	*A. mangium*	Shangsi County, Fangchenggang Region, Guangxi Province, China	22°06′50″ N, 107°52′60″ E	G.Q. Li	OQ659781	OQ659907	OQ660033
*P. kimberleyense*	AAA	CSF18957 ^e^	*P. massoniana*	Qiongshan District, Haikou Region, Hainan Province, China	19°40′39″ N, 110°26′51″ E	G.Q. Li	OQ659782	OQ659908	OQ660034
*P. kimberleyense*	AAA	CSF19124 ^e^	*A. mangium*	Ledong County, Hainan Province, China	18°44′44″ N, 109°13′43″ E	G.Q. Li	OQ659783	OQ659909	OQ660035
*P. kimberleyense*	AAA	CSF19126	*A. mangium*	Ledong County, Hainan Province, China	18°44′44″ N, 109°13′43″ E	G.Q. Li	OQ659784	OQ659910	OQ660036
*P. kimberleyense*	AAA	CSF19131	*A. mangium*	Ledong County, Hainan Province, China	18°44′44″ N, 109°13′43″ E	G.Q. Li	OQ659785	OQ659911	OQ660037
*P. kimberleyense*	AAA	CSF19134	*A. mangium*	Ledong County, Hainan Province, China	18°44′44″ N, 109°13′43″ E	G.Q. Li	OQ659786	OQ659912	OQ660038
*P. kimberleyense*	AAA	CSF19345	*A. mangium*	Huaan County, Zhangzhou Region, Fujian Province, China	24°57′47″ N, 117°31′40″ E	G.Q. Li	OQ659787	OQ659913	OQ660039
*P. kimberleyense*	AAA	CSF19348	*A. mangium*	Huaan County, Zhangzhou Region, Fujian Province, China	24°57′47″ N, 117°31′40″ E	G.Q. Li	OQ659788	OQ659914	OQ660040
*P. kimberleyense*	AAA	CSF19359	*A. mangium*	Huaan County, Zhangzhou Region, Fujian Province, China	24°57′47″ N, 117°31′40″ E	G.Q. Li	OQ659789	OQ659915	OQ660041
*P. kimberleyense*	AAA	CSF19659	*A. mangium*	Jiexi County, Jieyang Region, Guangdong Province, China	23°28′49″ N, 115°45′46″ E	G.Q. Li	OQ659790	OQ659916	OQ660042
*P. kimberleyense*	AAA	CSF19661	*A. mangium*	Jiexi County, Jieyang Region, Guangdong Province, China	23°28′49″ N, 115°45′46″ E	G.Q. Li	OQ659791	OQ659917	OQ660043
*P. kimberleyense*	AAB	CSF19094 ^d^	*A. mangium*	Ledong County, Hainan Province, China	18°44′44″ N, 109°13′43″ E	G.Q. Li	OQ659792	OQ659918	OQ660044
*P. kimberleyense*	AAB	CSF19099 ^d^	*A. mangium*	Ledong County, Hainan Province, China	18°44′44″N, 109°13′43″E	G.Q. Li	OQ659793	OQ659919	OQ660045
*P. kimberleyense*	AAB	CSF19106	*A. mangium*	Ledong County, Hainan Province, China	18°44′44″ N, 109°13′43″ E	G.Q. Li	OQ659794	OQ659920	OQ660046
*P. kimberleyense*	AAB	CSF19109	*A. mangium*	Ledong County, Hainan Province, China	18°44′44″ N, 109°13′43″ E	G.Q. Li	OQ659795	OQ659921	OQ660047
*P. kimberleyense*	AAB	CSF19111	*A. mangium*	Ledong County, Hainan Province, China	18°44′44″ N, 109°13′43″ E	G.Q. Li	OQ659796	OQ659922	OQ660048
*P. kimberleyense*	AAB	CSF19117	*A. mangium*	Ledong County, Hainan Province, China	18°44′44″ N, 109°13′43″ E	G.Q. Li	OQ659797	OQ659923	OQ660049
*P. kimberleyense*	AAB	CSF19120	*A. mangium*	Ledong County, Hainan Province, China	18°44′44″ N, 109°13′43″ E	G.Q. Li	OQ659798	OQ659924	OQ660050
*P. kimberleyense*	AAB	CSF19122	*A. mangium*	Ledong County, Hainan Province, China	18°44′44″ N, 109°13′43″ E	G.Q. Li	OQ659799	OQ659925	OQ660051
*P. kimberleyense*	AAB	CSF19129	*A. mangium*	Ledong County, Hainan Province, China	18°44′44″ N, 109°13′43″ E	G.Q. Li	OQ659800	OQ659926	OQ660052
*P. kimberleyense*	AAB	CSF19136	*A. mangium*	Ledong County, Hainan Province, China	18°44′44″ N, 109°13′43″E	G.Q. Li	OQ659801	OQ659927	OQ660053
*P. kimberleyense*	AAB	CSF19138	*A. mangium*	Ledong County, Hainan Province, China	18°44′44″N, 109°13′43″E	G.Q. Li	OQ659802	OQ659928	OQ660054
*P. kimberleyense*	AAC	CSF18423 ^d^	*P. massoniana*	Fengkai County, Zhaoqing Region, Guangdong Province, China	23°26′59″ N, 111°34′37″ E	G.Q. Li	OQ659803	OQ659929	OQ660055
*P. kimberleyense*	AAC	CSF18503 ^de^	*A. mangium*	Beiliu County, Yulin Region, Guangxi Province, China	22°47′12″ N, 110°17′53″ E	G.Q. Li	OQ659804	OQ659930	OQ660056
*P. kimberleyense*	AAC	CSF18517	*A. mangium*	Beiliu County, Yulin Region, Guangxi Province, China	22°47′12″ N, 110°17′53″ E	G.Q. Li	OQ659805	OQ659931	OQ660057
*P. kimberleyense*	AAD	CSF19107 ^d^	*A. mangium*	Ledong County, Hainan Province, China	18°44′44″ N, 109°13′43″ E	G.Q. Li	OQ659806	OQ659932	OQ660058
*P. kimberleyense*	ABA	CSF18642 ^d^	*P. massoniana*	Rongan County, Liuzhou Region, Guangxi Province, China	25°15′11″ N, 109°25′45″ E	G.Q. Li	OQ659807	OQ659933	OQ660059
*P. kimberleyense*	ABA	CSF18829 ^d^	*A. mangium*	Shangsi County, Fangchenggang Region, Guangxi Province, China	22°06′50″ N, 107°52′60″ E	G.Q. Li	OQ659808	OQ659934	OQ660060
*P. kimberleyense*	ABA	CSF18830	*A. mangium*	Shangsi County, Fangchenggang Region, Guangxi Province, China	22°06′50″ N, 107°52′60″ E	G.Q. Li	OQ659809	OQ659935	OQ660061
*P. kimberleyense*	ABA	CSF18842	*A. mangium*	Shangsi County, Fangchenggang Region, Guangxi Province, China	22°06′50″ N, 107°52′60″ E	G.Q. Li	OQ659810	OQ659936	OQ660062
*P. kimberleyense*	ABA	CSF18990	*A. mangium*	Qiongshan District, Haikou Region, Hainan Province, China	19°40′39″ N, 110°26′51″ E	G.Q. Li	OQ659811	OQ659937	OQ660063
*P. kimberleyense*	ABA	CSF18991	*A. mangium*	Qiongshan District, Haikou Region, Hainan Province, China	19°40′39″ N, 110°26′51″ E	G.Q. Li	OQ659812	OQ659938	OQ660064
*P. kimberleyense*	ABA	CSF19064 ^e^	*Eucalyptus* sp.	Ledong County, Hainan Province, China	18°44′44″ N, 109°13′43″ E	G.Q. Li	OQ659813	OQ659939	OQ660065
*P. kimberleyense*	ABA	CSF19067 ^e^	*Eucalyptus* sp.	Ledong County, Hainan Province, China	18°44′44″ N, 109°13′43″ E	G.Q. Li	OQ659814	OQ659940	OQ660066
*P. kimberleyense*	ABA	CSF19092	*A. mangium*	Ledong County, Hainan Province, China	18°44′44″ N, 109°13′43″ E	G.Q. Li	OQ659815	OQ659941	OQ660067
*P. kimberleyense*	ABA	CSF19116	*A. mangium*	Ledong County, Hainan Province, China	18°44′44″ N, 109°13′43″ E	G.Q. Li	OQ659816	OQ659942	OQ660068
*P. kimberleyense*	ABA	CSF19118	*A. mangium*	Ledong County, Hainan Province, China	18°44′44″ N, 109°13′43″ E	G.Q. Li	OQ659817	OQ659943	OQ660069
*P. kimberleyense*	ABA	CSF19123	*A. mangium*	Ledong County, Hainan Province, China	18°44′44″ N, 109°13′43″ E	G.Q. Li	OQ659818	OQ659944	OQ660070
*P. kimberleyense*	ABA	CSF19128	*A. mangium*	Ledong County, Hainan Province, China	18°44′44″ N, 109°13′43″ E	G.Q. Li	OQ659819	OQ659945	OQ660071
*P. kimberleyense*	ABC	CSF18375 ^d^	*P. massoniana*	Huazhou County, Maoming Region, Guangdong Province, China	21°47′05″ N, 110°28′35″ E	G.Q. Li	OQ659820	OQ659946	OQ660072
*P. kimberleyense*	ABD	CSF19112 ^d^	*A. mangium*	Ledong County, Hainan Province, China	18°44′44″ N, 109°13′43″ E	G.Q. Li	OQ659821	OQ659947	OQ660073
*P. kimberleyense*	ACA	CSF19125 ^d^	*A. mangium*	Ledong County, Hainan Province, China	18°44′44″ N, 109°13′43″ E	G.Q. Li	OQ659822	OQ659948	OQ660074
*P. kimberleyense*	ADB	CSF19093 ^d^	*A. mangium*	Ledong County, Hainan Province, China	18°44′44″ N, 109°13′43″ E	G.Q. Li	OQ659823	OQ659949	OQ660075
*P. kimberleyense*	AEE	CSF18839 ^d^	*A. mangium*	Shangsi County, Fangchenggang Region, Guangxi Province, China	22°06′50″ N, 107°52′60″ E	G.Q. Li	OQ659824	OQ659950	OQ660076
*P. kimberleyense*	AFA	CSF18961 ^d^	*A. mangium*	Qiongshan District, Haikou Region, Hainan Province, China	19°40′39″ N, 110°26′51″ E	G.Q. Li	OQ659825	OQ659951	OQ660077
*P. kimberleyense*	BAA	CSF14162 ^d^	*P. massoniana*	Gaozhou County, Maoming Region, Guangdong Province, China	22°11′31″ N, 110°44′45″ E	G.Q. Li	OQ659826	OQ659952	OQ660078
*P. kimberleyense*	BAA	CSF18376 ^de^	*P. massoniana*	Huazhou County, Maoming Region, Guangdong Province, China	21°47′05″ N, 110°28′35″ E	G.Q. Li	OQ659827	OQ659953	OQ660079
*P. kimberleyense*	BAA	CSF18407	*A. mangium*	Yangchun County, Yangjiang Region, Guangdong Province, China	21°55′32″ N, 111°38′39″ E	G.Q. Li	OQ659828	OQ659954	OQ660080
*P. kimberleyense*	BAA	CSF18425	*P. massoniana*	Fengkai County, Zhaoqing Region, Guangdong Province, China	23°26′59″ N, 111°34′37″ E	G.Q. Li	OQ659829	OQ659955	OQ660081
*P. kimberleyense*	BAA	CSF18491 ^e^	*P. massoniana*	Beiliu County, Yulin Region, Guangxi Province, China	22°47′12″ N, 110°17′53″ E	G.Q. Li	OQ659830	OQ659956	OQ660082
*P. kimberleyense*	BAA	CSF18522	*A. mangium*	Beiliu County, Yulin Region, Guangxi Province, China	22°47′12″ N, 110°17′53″E	G.Q. Li	OQ659831	OQ659957	OQ660083
*P. kimberleyense*	BAA	CSF18538	*A. mangium*	Beiliu County, Yulin Region, Guangxi Province, China	22°47′12″ N, 110°17′53″ E	G.Q. Li	OQ659832	OQ659958	OQ660084
*P. kimberleyense*	BAA	CSF18539	*A. mangium*	Beiliu County, Yulin Region, Guangxi Province, China	22°47′12″ N, 110°17′53″ E	G.Q. Li	OQ659833	OQ659959	OQ660085
*P. kimberleyense*	BAA	CSF18822	*A. mangium*	Shangsi County, Fangchenggang Region, Guangxi Province, China	22°06′50″ N, 107°52′60″ E	G.Q. Li	OQ659834	OQ659960	OQ660086
*P. kimberleyense*	BAA	CSF18823	*A. mangium*	Shangsi County, Fangchenggang Region, Guangxi Province, China	22°06′50″ N, 107°52′60″ E	G.Q. Li	OQ659835	OQ659961	OQ660087
*P. kimberleyense*	BAA	CSF18825	*A. mangium*	Shangsi County, Fangchenggang Region, Guangxi Province, China	22°06′50″N, 107°52′60″E	G.Q. Li	OQ659836	OQ659962	OQ660088
*P. kimberleyense*	BAA	CSF18919	*P. massoniana*	Qiongshan District, Haikou Region, Hainan Province, China	19°40′39″ N, 110°26′51″ E	G.Q. Li	OQ659837	OQ659963	OQ660089
*P. kimberleyense*	BAA	CSF18923	*P. massoniana*	Qiongshan District, Haikou Region, Hainan Province, China	19°40′39″ N, 110°26′51″ E	G.Q. Li	OQ659838	OQ659964	OQ660090
*P. kimberleyense*	BAA	CSF18924	*P. massoniana*	Qiongshan District, Haikou Region, Hainan Province, China	19°40′39″ N, 110°26′51″ E	G.Q. Li	OQ659839	OQ659965	OQ660091
*P. kimberleyense*	BAA	CSF18973	*A. mangium*	Qiongshan District, Haikou Region, Hainan Province, China	19°40′39″ N, 110°26′51″ E	G.Q. Li	OQ659840	OQ659966	OQ660092
*P. kimberleyense*	BAA	CSF18993	*A. mangium*	Qiongshan District, Haikou Region, Hainan Province, China	19°40′39″ N, 110°26′51″ E	G.Q. Li	OQ659841	OQ659967	OQ660093
*P. kimberleyense*	BAA	CSF19135	*A. mangium*	Ledong County, Hainan Province, China	18°44′44″ N, 109°13′43″ E	G.Q. Li	OQ659842	OQ659968	OQ660094
*P. kimberleyense*	BAA	CSF19182	*A. mangium*	Danzhou Region, Hainan Province, China	19°41′42″ N, 109°19′50″ E	G.Q. Li	OQ659843	OQ659969	OQ660095
*P. kimberleyense*	BAA	CSF19318 ^e^	*A. mangium*	Huaan County, Zhangzhou Region, Fujian Province, China	24°46′43″ N, 117°36′11′ ’E	G.Q. Li	OQ659844	OQ659970	OQ660096
*P. kimberleyense*	BAA	CSF19324	*A. mangium*	Huaan County, Zhangzhou Region, Fujian Province, China	24°46′43″ N, 117°36′11″ E	G.Q. Li	OQ659845	OQ659971	OQ660097
*P. kimberleyense*	BAA	CSF19325	*A. mangium*	Huaan County, Zhangzhou Region, Fujian Province, China	24°46′43″ N, 117°36′11″ E	G.Q. Li	OQ659846	OQ659972	OQ660098
*P. kimberleyense*	BAA	CSF19327	*A. mangium*	Huaan County, Zhangzhou Region, Fujian Province, China	24°57′47″ N, 117°31′40″ E	G.Q. Li	OQ659847	OQ659973	OQ660099
*P. kimberleyense*	BAA	CSF19328	*A. mangium*	Huaan County, Zhangzhou Region, Fujian Province, China	24°57′47″ N, 117°31′40″ E	G.Q. Li	OQ659848	OQ659974	OQ660100
*P. kimberleyense*	BAA	CSF19336	*A. mangium*	Huaan County, Zhangzhou Region, Fujian Province, China	24°57′47″ N, 117°31′40″ E	G.Q. Li	OQ659849	OQ659975	OQ660101
*P. kimberleyense*	BAA	CSF19337	*A. mangium*	Huaan County, Zhangzhou Region, Fujian Province, China	24°57′47″ N, 117°31′40″ E	G.Q. Li	OQ659850	OQ659976	OQ660102
*P. kimberleyense*	BAA	CSF19341	*A. mangium*	Huaan County, Zhangzhou Region, Fujian Province, China	24°57′47″ N, 117°31′40″ E	G.Q. Li	OQ659851	OQ659977	OQ660103
*P. kimberleyense*	BAA	CSF19353	*A. mangium*	Huaan County, Zhangzhou Region, Fujian Province, China	24°57′47″ N, 117°31′40″ E	G.Q. Li	OQ659852	OQ659978	OQ660104
*P. kimberleyense*	BAA	CSF19354	*A. mangium*	Huaan County, Zhangzhou Region, Fujian Province, China	24°57′47″N, 117°31′40″E	G.Q. Li	OQ659853	OQ659979	OQ660105
*P. kimberleyense*	BAA	CSF19358	*A. mangium*	Huaan County, Zhangzhou Region, Fujian Province, China	24°57′47″ N, 117°31′40″ E	G.Q. Li	OQ659854	OQ659980	OQ660106
*P. kimberleyense*	BAA	CSF19650	*A. mangium*	Jiexi County, Jieyang Region, Guangdong Province, China	23°28′49″ N, 115°45′46″ E	G.Q. Li	OQ659855	OQ659981	OQ660107
*P. kimberleyense*	BAA	CSF19658	*A. mangium*	Jiexi County, Jieyang Region, Guangdong Province, China	23°28′49″ N, 115°45′46″ E	G.Q. Li	OQ659856	OQ659982	OQ660108
*P. kimberleyense*	BAC	CSF19110 ^d^	*A. mangium*	Ledong County, Hainan Province, China	18°44′44″ N, 109°13′43″ E	G.Q. Li	OQ659857	OQ659983	OQ660109
*P. kimberleyense*	BAC	CSF19649 ^d^	*A. mangium*	Jiexi County, Jieyang Region, Guangdong Province, China	23°28′49″ N, 115°45′46″ E	G.Q. Li	OQ659858	OQ659984	OQ660110
*P. kimberleyense*	BBA	CSF14610 ^d^	*A. mangium*	Yangdong County, Yangjiang Region, Guangdong Province, China	22°01′27″N, 112°11′17″ E	G.Q. Li	OQ659859	OQ659985	OQ660111
*P. kimberleyense*	BBA	CSF18513	*A. mangium*	Beiliu County, Yulin Region, Guangxi Province, China	22°47′12″ N, 110°17′53″ E	G.Q. Li	OQ659860	OQ659986	OQ660112
*P. kimberleyense*	BBA	CSF18835	*A. mangium*	Shangsi County, Fangchenggang Region, Guangxi Province, China	22°06′50″ N, 107°52′60″ E	G.Q. Li	OQ659861	OQ659987	OQ660113
*P. kimberleyense*	BBA	CSF18854	*A. mangium*	Shangsi County, Fangchenggang Region, Guangxi Province, China	22°06′50″ N, 107°52′60″ E	G.Q. Li	OQ659862	OQ659988	OQ660114
*P. kimberleyense*	BBA	CSF18858	*A. mangium*	Shangsi County, Fangchenggang Region, Guangxi Province, China	22°06′50″ N, 107°52′60″ E	G.Q. Li	OQ659863	OQ659989	OQ660115
*P. kimberleyense*	BBA	CSF19653 ^d^	*A. mangium*	Jiexi County, Jieyang Region, Guangdong Province, China	23°28′49″N, 115°45′46″E	G.Q. Li	OQ659864	OQ659990	OQ660116
*P. kimberleyense*	BCA	CSF19121 ^d^	*A. mangium*	Ledong County, Hainan Province, China	18°44′44″ N, 109°13′43″ E	G.Q. Li	OQ659865	OQ659991	OQ660117
*P. kimberleyense*	BCA	CSF19137 ^d^	*A. mangium*	Ledong County, Hainan Province, China	18°44′44″ N, 109°13′43″ E	G.Q. Li	OQ659866	OQ659992	OQ660118
*P. kimberleyense*	CAA	CSF19096 ^d^	*A. mangium*	Ledong County, Hainan Province, China	18°44′44″ N, 109°13′43″ E	G.Q. Li	OQ659867	OQ659993	OQ660119
*P. kimberleyense*	CBA	CSF19066 ^de^	*Eucalyptus* sp.	Ledong County, Hainan Province, China	18°44′44″ N, 109°13′43″ E	G.Q. Li	OQ659868	OQ659994	OQ660120
*P. kimberleyense*	DBA	CSF19343 ^d^	*A. mangium*	Huaan County, Zhangzhou Region, Fujian Province, China	24°57′47″ N, 117°31′40″ E	G.Q. Li	OQ659869	OQ659995	OQ660121
*P. violaceum*	AAA	CSF18527 ^d^	*A. mangium*	Beiliu County, Yulin Region, Guangxi Province, China	22°47′12″ N, 110°17′53″ E	G.Q. Li	OQ659870	OQ659996	OQ660122
*P. violaceum*	AAA	CSF18841 ^d^	*A. mangium*	Shangsi County, Fangchenggang Region, Guangxi Province, China	22°06′50″ N, 107°52′60″ E	G.Q. Li	OQ659871	OQ659997	OQ660123
*P. violaceum*	AAA	CSF19180	*A. mangium*	Danzhou Region, Hainan Province, China	19°41′42″ N, 109°19′50″ E	G.Q. Li	OQ659872	OQ659998	OQ660124
*P. violaceum*	AAA	CSF19339	*A. mangium*	Huaan County, Zhangzhou Region, Fujian Province, China	24°57′47″ N, 117°31′40″ E	G.Q. Li	OQ659873	OQ659999	OQ660125
*P. violaceum*	AAB	CSF19320 ^de^	*A. mangium*	Huaan County, Zhangzhou Region, Fujian Province, China	24°46′43″ N, 117°36′11″ E	G.Q. Li	OQ659874	OQ660000	OQ660126
*P. violaceum*	AAB	CSF19321 ^d^	*A. mangium*	Huaan County, Zhangzhou Region, Fujian Province, China	24°46′43″ N, 117°36′11″ E	G.Q. Li	OQ659875	OQ660001	OQ660127
*P. violaceum*	AAB	CSF19323	*A. mangium*	Huaan County, Zhangzhou Region, Fujian Province, China	24°46′43″ N, 117°36′11″ E	G.Q. Li	OQ659876	OQ660002	OQ660128
*P. violaceum*	ABA	CSF19259 ^de^	*P. massoniana*	Huaan County, Zhangzhou Region, Fujian Province, China	24°46′43″ N, 117°36′11″ E	G.Q. Li	OQ659877	OQ660003	OQ660129
*P. violaceum*	ABA	CSF19335 ^d^	*A. mangium*	Huaan County, Zhangzhou Region, Fujian Province, China	24°57′47″ N, 117°31′40″ E	G.Q. Li	OQ659878	OQ660004	OQ660130
*P. violaceum*	ABA	CSF19338	*A. mangium*	Huaan County, Zhangzhou Region, Fujian Province, China	24°57′47″ N, 117°31′40″ E	G.Q. Li	OQ659879	OQ660005	OQ660131
*P. violaceum*	ABA	CSF19418 ^e^	*P. massoniana*	Yongtai County, Fuzhou Region, Fujian Province, China	25°54′02″ N, 118°54′50″ E	G.Q. Li	OQ659880	OQ660006	OQ660132
*P. violaceum*	ABA	CSF19419	*P. massoniana*	Yongtai County, Fuzhou Region, Fujian Province, China	25°54′02″ N, 118°54′50″ E	G.Q. Li	OQ659881	OQ660007	OQ660133
*P. violaceum*	ACA	CSF18515 ^d^	*A. mangium*	Beiliu County, Yulin Region, Guangxi Province, China	22°47′12″ N, 110°17′53″ E	G.Q. Li	OQ659882	OQ660008	OQ660134
*P. violaceum*	ACA	CSF19357 ^d^	*A. mangium*	Huaan County, Zhangzhou Region, Fujian Province, China	24°57′47″ N, 117°31′40″ E	G.Q. Li	OQ659883	OQ660009	OQ660135
*P. violaceum*	ADA	CSF18979 ^d^	*A. mangium*	Qiongshan District, Haikou Region, Hainan Province, China	19°40′39″ N, 110°26′51″ E	G.Q. Li	OQ659884	OQ660010	OQ660136
*P. violaceum*	ADB	CSF18972 ^d^	*A. mangium*	Qiongshan District, Haikou Region, Hainan Province, China	19°40′39″ N, 110°26′51″ E	G.Q. Li	OQ659885	OQ660011	OQ660137
*P. violaceum*	AEA	CSF18430 ^de^	*P. massoniana*	Ruyuan County, Shaoguan Region, Guangdong Province, China	24°50′13″ N, 113°21′03″ E	G.Q. Li	OQ659886	OQ660012	OQ660138
*P. violaceum*	BAA	CSF18827 ^de^	*A. mangium*	Shangsi County, Fangchenggang Region, Guangxi Province, China	22°06′50″ N, 107°52′60″ E	G.Q. Li	OQ659887	OQ660013	OQ660139
*P. violaceum*	BAA	CSF18828 ^d^	*A. mangium*	Shangsi County, Fangchenggang Region, Guangxi Province, China	22°06′50″ N, 107°52′60″ E	G.Q. Li	OQ659888	OQ660014	OQ660140
*P. violaceum*	BAA	CSF18844	*A. mangium*	Shangsi County, Fangchenggang Region, Guangxi Province, China	22°06′50″ N, 107°52′60″ E	G.Q. Li	OQ659889	OQ660015	OQ660141
*P. violaceum*	BAA	CSF18895 ^e^	*Eucalyptus* sp.	Qiongshan District, Haikou Region, Hainan Province, China	19°40′39″ N, 110°26′51″ E	G.Q. Li	OQ659890	OQ660016	OQ660142
*P. violaceum*	BAA	CSF19222	*A. mangium*	Danzhou Region, Hainan Province, China	19°41′42″ N, 109°19′50″ E	G.Q. Li	OQ659891	OQ660017	OQ660143
*P. violaceum*	BAA	CSF19351	*A. mangium*	Huaan County, Zhangzhou Region, Fujian Province, China	24°57′47″ N, 117°31′40″ E	G.Q. Li	OQ659892	OQ660018	OQ660144
*P. violaceum*	BAB	CSF18533 ^d^	*A. mangium*	Beiliu County, Yulin Region, Guangxi Province, China	22°47′12″ N, 110°17′53″ E	G.Q. Li	OQ659893	OQ660019	OQ660145
*P. violaceum*	BAC	CSF18840 ^d^	*A. mangium*	Shangsi County, Fangchenggang Region, Guangxi Province, China	22°06′50″ N, 107°52′60″ E	G.Q. Li	OQ659894	OQ660020	OQ660146
*P. violaceum*	BBA	CSF18859 ^d^	*A. mangium*	Shangsi County, Fangchenggang Region, Guangxi Province, China	22°06′50″ N, 107°52′60″ E	G.Q. Li	OQ659895	OQ660021	OQ660147
*P. violaceum*	BCA	CSF19344 ^d^	*A. mangium*	Huaan County, Zhangzhou Region, Fujian Province, China	24°57′47″ N, 117°31′40″ E	G.Q. Li	OQ659896	OQ660022	OQ660148
*P. violaceum*	BCB	CSF18843 ^d^	*A. mangium*	Shangsi County, Fangchenggang Region, Guangxi Province, China	22°06′50″ N, 107°52′60″ E	G.Q. Li	OQ659897	OQ660023	OQ660149
*P. violaceum*	BDA	CSF19202 ^d^	*A. mangium*	Danzhou Region, Hainan Province, China	19°41′42″ N, 109°19′50″ E	G.Q. Li	OQ659898	OQ660024	OQ660150
*P. violaceum*	BDB	CSF19217 ^de^	*A. mangium*	Danzhou Region, Hainan Province, China	19°41′42″ N, 109°19′50′′ E	G.Q. Li	OQ659899	OQ660025	OQ660151
*P. violaceum*	CAA	CSF19347 ^d^	*A. mangium*	Huaan County, Zhangzhou Region, Fujian Province, China	24°57′47′′ N, 117°31′40′′ E	G.Q. Li	OQ659900	OQ660026	OQ660152

^a^ Genotype within each species determined by ITS, *tef1*, and *tub2* loci. The three capital letters of genotype represent the ITS, *tef1*, and *tub2* sequences, respectively. The same letter among isolates from each species means they shared the same genotype. ^b^ CSF: Culture Collection of the Research Institute of Fast-growing trees, Chinese Academy of Forestry, Zhanjiang, Guangdong Province, China. ^c^ ITS: internal transcribed spacer; *tef1*: translation elongation factor 1-alpha; *tub2*: β-tubulin. ^d^ Isolates used for phylogenetic analyses. ^e^ Isolates used for inoculation trials.

**Table 3 pathogens-12-00574-t003:** Isolates from other studies and used for phylogenetic analyses for this study.

Species	Isolate No.^a^	Host	Location	Collector	GenBank Accession No.^b^			Reference
					ITS	*tef1*	*tub2*	
*Pseudofusicoccum adansoniae*	**CMW 26147 = CBS 122055**	*Adansonia gibbosa*	Australia	T.I. Burgess	EF585523	EF585571	MT592771	[5,32]
*P. adansoniae*	CMW26146 = CBS 122054	*Eucalyptus* sp.	Australia	T.I. Burgess	EF585532	EF585570	MT592770	[5,32]
*P. africanum*	**CMW 48028 = PPRI 25471**	*Mimusops caffra*	South Africa	M.J. Wingfield	MH558614	MH576590	NA	[33]
*P. africanum*	CMW 48027	*Mimusops caffra*	South Africa	M.J. Wingfield	MH558616	MH576591	NA	[33]
*P. ardesiacum*	**CMW 26159 = CBS 122062**	*Adansonia gibbosa*	Australia	T.I. Burgess	EU144060	EU144075	KX465069	[3,32]
*P. ardesiacum*	CMW 26155 = CBS 122063	*Adansonia gibbosa*	Australia	T.I. Burgess	EU144061	EU144076	KX465070	[3,32]
*P. artocarpi*	**CPC 22796 = CBS 138655**	*Artocarpus heterophyllus*	Thailand	T. Trakunyingcharoen	KM006452	KM006483	MT882262	[5,34]
*P. calophylli*	**MFLUCC 17-2533 = KUMCC 18-0282**	*Calophyllum inophyllum*	Thailand	S.C. Jayasiri	MK347764	MK340877	MK412885	[35]
*P. kimberleyensis*	**CMW 26156 = CBS 122058**	*Acacia synchronica*	Australia	T.I. Burgess	EU144057	EU144072	MT592773	[5,32]
*P. kimberleyensis*	CMW 26157 = CBS 122059	*Eucalyptus* sp.	Australia	T.I. Burgess	EU144056	EU144071	MT592774	[5,32]
*P. olivaceum*	**CMW 20881 = CBS 124939**	*Pterocarpus angolensis*	South Africa	J. Roux	FJ888459	FJ888437	MT592776	[5,36]
*P. olivaceum*	CMW 22637 = CBS 124940	*Pterocarpus angolensis*	South Africa	J. Mehl & J. Roux	FJ888462	FJ888438	MT592777	[5,36]
*P. stromaticum*	**CMW 13434 = CBS 117448**	*Eucalyptus* hybrid	Venezuela	S. Mohali	AY693974	AY693975	EU673094	[2,37]
*P. stromaticum*	CMW 13435 = CBS 117449	*Eucalyptus* hybrid	Venezuela	S. Mohali	DQ436935	DQ436936	EU673093	[2,37]
*P. violaceum*	**CMW 22679 = CBS 124936**	*Pterocarpus angolensis*	South Africa	J. Mehl & J. Roux	FJ888474	FJ888442	MT592782	[5,36]
*P. violaceum*	CMW 20436 = CBS 124937	*Pterocarpus angolensis*	South Africa	J. Roux	FJ888458	FJ888440	MT592783	[5,36]
*B. dothidea*	**CBS 115476 = CMW 8000**	*Prunus* sp.	Switzerland	B. Slippers	AY236949	AY236898	AY236927	[38]

^a^ Isolates in bold represent ex-type. CBS: Westerdijk Fungal Biodiversity Institute, Utrecht, The Netherlands; CMW: Culture collection of the Forestry and Agricultural Biotechnology Institute, University of Pretoria, Pretoria, South Africa; CPC = Culture Collection of P.W. Crous, housed at CBS; KUMCC: Kunming Institute of Botany Culture Collection, Yunnan, China; MFLUCC: Mae Fah Luang University Culture Collection, Chiang Rai, Thailand; PPRI: the South African National Collection of Fungi, Roodeplaat, South Africa. ^b^ ITS: internal transcribed spacer; *tef1*: translation elongation factor 1-alpha; *tub2*: β-tubulin.

## Data Availability

The DNA sequences generated in this paper were submitted to the NCBI database (https://www.ncbi.nlm.nih.gov/genbank/, accession numbers listed in the Table 2, last accessed on 22 March 2023).

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
