# Peer review of "Characterization of Pseudofusicoccum Species from Diseased Plantation-Grown Acacia mangium, Eucalyptus spp., and Pinus massoniana in Southern China"

_pathogens, 2023, doi:10.3390/pathogens12040574_

Round 1

Reviewer 1 Report

The paper is well-analyzed, and the grammar and writing are good too. Check the positions of some tables since I think some should be placed in the results. 

Reviewer 2 Report

The survey that is described in this work is the result of a huge amount of analyses that include pathogen isolation, molecular characterization and pathogenicity tests. However the description of the isolated strains is very pour: it is suggested to describe the colonies and the microscopic features. The method and the results should be described with more details. 

Some other comments follow.

Line 19: What do you mean with “reminder”?

Line 22: change “obvious” with “expected”

Line 53: delete “these”

Line 68: Briefly describe the method of isolation used

Line 102. Table 2: Genbank accession number. Also provide the percentage of similarity of the isolate from this study with those retrieved from Genbank. Provide more explanation about genotypes.

Line 161: “all 126 isolates”

Line 163. Please, complete the phrase. Replace XXXX with the definitive text

Figure 3. The difference among negative controls and inoculated plants is very low, at least in the picture provided for Pinus elliottii and (mainly) on Eucalyptus grandis × E. urophylla. It is suggested to add a description of the symptoms to allow the readers to verify what the authors are trying to demonstrate and to confirm the result of the statistical analysis.

Round 2

Reviewer 2 Report

The manuscript results sustantially improved. It is suggested to review English editing. As an example, two correction in text follows:

Line 19: the resto the isolate

Line 78: With the CTAB method